# Effects of Transportation on Blood Indices, Oxidative Stress, Rumen Fermentation Parameters and Rumen Microbiota in Goats

**DOI:** 10.3390/ani14111616

**Published:** 2024-05-29

**Authors:** Rui Li, Lizhi Wang, Binlong Chen, Yi Zhang, Pei Qi

**Affiliations:** 1Animal Nutrition Institute, Sichuan Agricultural University, Chengdu 611130, China; 15517887509@163.com (R.L.); qipei@stu.sicau.edu.cn (P.Q.); 2Key Laboratory of Local Characteristic Goat, Xi Chang University, Xichang 615012, China; binlong2369@163.com

**Keywords:** goats, transport stress, rumen fermentation, rumen microbiota

## Abstract

**Simple Summary:**

Transportation is an inevitable part of the raising process of goats. During transportation, animals are subjected to ‘transport stress’ due to limited space, bumpy roads, lack of feed and drinking water, and other stimuli. In order to gain a comprehensive understanding of how transport stress affects goats, we conducted our experimental studies. Sixteen healthy goats were selected, and these goats were transported at a speed ranging from 35 to 45 km/h for 20 h. We measured changes in the physiological indexes, blood physiological indexes, biochemical indexes, rumen fermentation indexes, and rumen microbial structure composition of goats before and after transportation. Our experimental findings revealed that continuous transportation for a duration of 20 h can induce a severe stress response in goats, leading to compromised immune function, diminished antioxidant capacity, escalated inflammatory response, and altered rumen fermentation indices. However, they did not reveal any significant impact on the structure and composition of the rumen microbiota.

**Abstract:**

The objective of this experiment was to delve into the impacts of transportation on goats. Sixteen healthy goats were selected as experimental animals; these goats were transported at a speed ranging from 35 to 45 km/h for 20 h. The changes in the physiological indexes, blood physiological indexes, biochemical indexes, rumen fermentation indexes, and rumen microbial structure composition of goats before and after transportation were measured. The results showed that after transportation, the contents of IgM, IgA, IgG, and Thyroxine decreased very significantly, while the contents of propionic acid, Hemoglobin and Epinephrine significantly increased, and the contents of VFA, acetic acid, butyric acid, isobutyric acid, isovaleric acid, LPS, IL-1β, IL-6, TNF-α, Major Acute Phase Protein, protein carbonyl, and cortisol increased very significantly. There was no significant difference in α-diversity and β-diversity, and the relative abundance of rumen microorganisms was not significantly different at either phylum or genus levels. The experimental findings revealed that continuous transportation for a duration of 20 h can induce a severe stress response in goats, leading to compromised immune function, diminished antioxidant capacity, escalated inflammatory response, and altered rumen fermentation indices. However, the experiment did not reveal any significant impact on the structure and composition of the rumen microbiota.

## 1. Introduction

In the raising of goats, transportation is an inevitable part due to the need for breeding, fattening, and marketing. During transportation, due to the narrow space, bumpy roads [1,2], restriction of feed and drinking water, and other stimuli [3], animals will suffer from ‘transport stress’ [4], a series of uncomfortable manifestations such as listlessness, loss of appetite, dyspnea, diarrhea and elevated body temperature, which leads to digestive and metabolic disorders, reduced production performance, imbalance of immune function, decreased disease resistance [5], and in severe cases, even animal death, thus causing serious economic losses to farmers. The adverse state of animals subjected to transport stress can be evaluated by observing their behavior and measuring changes in physiological indicators and certain blood biochemical indicators [6]. It was found that the concentration of the pro-inflammatory cytokines TNF-α, IL-1β, and IL-6 in the blood of animals increased significantly after transportation, and the inflammatory response of animals expanded, which resulted in damage to the body [7]. The damage caused by transport stress may be related to the disorder of rumen microorganisms, because rumen microorganisms are closely related to the health of the host [8]. Previous studies have found that transport leads to significant changes in the structure and composition of rumen microorganisms in beef cattle [7]. However, there are few such studies, especially on the effects of transport stress on the rumen microbiota of goats. To gain a comprehensive understanding of how transport stress affects goats and to unravel the underlying mechanisms of the damage it causes, we conducted the following experimental studies.

## 2. Materials and Methods

### 2.1. Animals, Feed, and Management

The experimental subjects were sixteen healthy goats, with an average weight of 30.2 ± 1.5 kg, housed at the Animal Experiment Base of Sichuan Agricultural University. The breed of goat was Jianzhou big-eared goat. During the experimental period, each goat was housed separately and fed a diet consisting of 50% corn, 14% soybean meal, 30% alfalfa hay, 0.5% NaCl, 1.5% baking soda, and 4% premix (containing minerals and vitamins).This diet was administered regularly twice daily at 9:00 a.m. and 5:00 p.m., with unlimited access to drinking water. The entire experiment spanned 18 days. Initially, a feeding trial was conducted for 16 days. On the 17th day, measurements of respiratory rate, pulse rate, and body temperature were taken at 8:00 a.m., followed by the collection of blood samples and rumen content samples. Subsequently, the experimental goats were transported by road at a speed ranging from 35 to 45 km/h from the Animal Experiment Base of Sichuan Agricultural University to Panzhihua City. From Panzhihua City, they were then returned to the Ya’an test base without any rest. The goats underwent a total journey of 800 km with a transport duration of 20 h. The transport vehicle provided a loading density of 1.5 m^2^ per goat. Weather conditions were favorable during transportation, maintaining an interior vehicle temperature of approximately 25 to 30 °C. Food and water were withheld during the transport period. Upon returning to the test base on the 18th day of the experiment, at approximately 4:00 a.m., the experimental animals were allowed to rest for 1 h before the water supply was restricted. After an additional 3 h, water was made freely available. Subsequently, measurements of respiratory rate, pulse rate, and body temperature were conducted, followed by the collection of blood and rumen content samples once again.

### 2.2. Samples Collection

#### 2.2.1. Collection of Rumen Content Samples

On the mornings of the 17th and 18th experimental days, the rumen contents of the test goats were collected using a gastric tube connected to a vacuum pump. The initial 50 mL of content was discarded, and then 60 mL was collected. Immediately following collection, the pH value of the sample was measured. Subsequently, the rumen content was separated into two portions. The first 50 mL portion was immediately added with metaphosphoric acid after sampling and then transferred to a nitrogen-filled gas bag and placed on ice. To ensure that the solid-phase microorganisms were fully dispersed into the liquid phase, the sample was repeatedly tapped. This liquid was then filtered through four layers of gauze into a chilled beaker, yielding rumen liquor, which was stored at −20 °C for future use. The second portion was divided into sterile 10 mL centrifuge tubes, wrapped in aluminum foil, and quickly frozen in liquid nitrogen. Immediately after sampling, these tubes were transferred to a −80 °C ultra-low-temperature freezer for long-term storage. These samples were later used for the analysis of microbial structural composition.

#### 2.2.2. Collection of Blood Samples

On the mornings of the 17th and 18th days of the experiment, three tubes of jugular vein blood samples were collected using 5 mL vacuum blood collection tubes. Two of these tubes were placed in regular blood collection tubes without any anticoagulants. After allowing the blood to clot for 45 min at room temperature, they were centrifuged at 3000 r/min for 10 min at 4 °C. The upper layer of serum was carefully extracted and stored in a freezer at −20 °C for future use. The third tube was placed in a blood collection tube containing EDTA for anticoagulation and then refrigerated at 4 °C, preserving it as a whole blood sample for the subsequent detection of routine blood indicators.

### 2.3. Indicator Measurement

#### 2.3.1. Physiological Indicators

To determine the respiratory frequency of the goats, observations were made of the chest and abdomen movements while they were in a resting state. Each rise and fall cycle of the chest or abdomen was counted as one breath. Using a stopwatch and a counter, the number of breaths was recorded over a period of 3 min to calculate the respiratory frequency (expressed as times per minute).

Following the respiratory rate measurement, the rectal temperature of each goat was determined using a veterinary thermometer. Standing directly behind the goat, one hand firmly grasped the tail and lifted it to expose the anus fully. On the other hand, the mercury column of the thermometer was lowered to below 35 °C, disinfected, and lightly lubricated. The thermometer was then carefully inserted into the rectum, rotated gently to advance it to more than 4/5 of its length, and clipped onto the coat at the base of the tail. After waiting for 3–5 min, the thermometer was removed, and the temperature indicated by the mercury column was noted. Immediately after measuring the body temperature, a stethoscope was placed over the heart area of the goat to auscultate the heartbeats. The number of heartbeats was counted over a one-minute period to record the pulse rate of the animal.

#### 2.3.2. Rumen fermentation parameter detection

(1)Rumen fluid pH: This was measured by a Remco portable acidimeter (PHBJ-260, Shanghai, China).(2)Concentration of Microbial Crude Protein (MCP) in rumen fluid: The MCP content in rumen fluid was determined by ‘differential centrifugation’ using a BCA protein quantitative kit. The kit was purchased from the Thermo Fisher Scientific (Waltham, MA, USA) and the specific operation steps referred to the kit instructions.(3)Concentration of short-chain fatty acid (SCFA) in rumen fluid: The rumen fluid underwent centrifugation at 4000 r/min for 10 min. Subsequently, 1 mL of the resulting supernatant was transferred to a 1.5 mL centrifuge tube containing 0.2 mL of 25% metaphosphoric acid for deproteinization. The mixture was thoroughly blended and chilled on ice for 30 min. Following this, the sample was centrifuged again at 10,000 r/min for 10 min. The supernatant was then aspirated using a disposable syringe and filtered through a 0.22 μM filter membrane into a 2 mL brown sample bottle. The concentration of SCFA was ascertained by gas chromatography (CP-3800, Varian, Santa Clara, CA, USA).

#### 2.3.3. Determination of Blood Samples

(1)Blood routine indexes: Whole blood samples were sent to the Laboratory Department of Ya’an People’s Hospital to measure blood routine indexes with a routine blood detector.(2)Serum antioxidant indexes: The activities of T-AOC, CAT, SOD, and GSH-Px, as well as the MDA content, were determined using kits obtained from Beijing Solarbio Science & Technology Co., Ltd. (Beijing, China). The measurements were conducted in accordance with the detailed instructions provided within each kit.(3)Serum immune indexes: The concentrations of TNF-α, IL-1β, IL-6, IL-10, and IL-4 in the serum samples were measured using an enzyme-linked immunosorbent assay (ELISA) kit purchased from Jiangsu Meimian Industrial Co., Ltd. (Kunshan, China). The specific operational steps followed the instructions provided in the kit.(4)Serum biochemical indexes: The levels of ALT, BUN, TP, ALB, TG, Glu, and lipoproteins (LDL-C, HDL-C) in serum were determined using a fully automatic biochemical analyzer (Hitachi 3100, Tokyo, Japan) in conjunction with standard kits obtained from Biosino Bio-Technology and Science Incorporation. The measurements were conducted following the instructions provided in the respective kits.(5)Serum hormonal indices: The concentrations of T4, COR, and AD in serum were assayed using an enzyme-linked immunosorbent assay (ELISA) kit sourced from Jiangsu Meimian Industrial Co., Ltd. The ELISA procedure was carried out according to the specific operation steps detailed in the kit’s instructions.

#### 2.3.4. Determination of Rumen Microbial Structure and Composition

The V4 hypervariable region of bacterial 16S rRNA was amplified via PCR, and the resulting products were analyzed through agarose gel electrophoresis at a 2% concentration. Aliquots were then mixed based on the PCR product concentration, and the target bands were recovered using Qiagens Gel Recovery Kit (Qiagen Company Limited, Shenzhen, China).Subsequently, the library was constructed and quantified with the NEBNext^®^ Ultra™ IIDNA Library Prep Kit (New England Biolabs Ltd., Beijing, China). Upon passing quality control, online sequencing was performed using NovaSeq6000.

The downloaded data were split into individual samples, and FLASH (V1.2.11) software was utilized to splice the reads of each sample, generating Raw Tags. Subsequently, fastp software (V0.23.1) was employed for quality control measures, resulting in Clean Tags. The Clean Tags were then compared to a database using Usearch software (v8.1.1756, http://www.drive5.com/usearch/, accessed on 1 April 2024) to obtain the final Effective Tags. Following this, QIIME2 software (https://qiime2.org/, accessed on 1 April 2024) was utilized to filter out sequences with an abundance of less than 5, yielding the final ASVs (Amplicon Sequence Variants) and characteristic tables. QIIME2 was further used to compare these ASVs against its database, thereby obtaining species information for each ASV.

The DADA2 method was used to deduplicate, and each deduplicated sequence was called an ASV (Amplicon Sequence Variant). According to the results of ASVs obtained by noise reduction and the research needs, the common and unique ASVs between different groups were analyzed. Most of the ASVs obtained were shared in each group.

The alpha diversity index was calculated using QIIME2 software. In cases where groups were present, an analysis of differences in alpha diversity between these groups was conducted. Additionally, Unifrac distances were computed using QIIME2. To assess significant differences in community structure between groups, the adonis and anosim functions within QIIME2 were employed. Lastly, R software (V02-0. 2010) facilitated the analysis of significantly differing species amongst the groups.

In the investigation of Beta diversity, the weighted UniFrac metric was used to assess the dissimilarity coefficient between two samples. Additionally, principal coordinate analysis (PCoA) was utilized to scrutinize the similarity among various samples. The underlying principle is that samples with closer proximity exhibit a more analogous species composition structure.

### 2.4. Statistical Analysis

Excel (2019) was initially utilized to organize and preprocess the data, followed by analysis using SPSS 27.0 statistical software to determine any variations in the test indices prior to and after transportation. Data adhering to the normal distribution underwent the independent sample *t*-test, while the non-parametric *t*-test was employed for data deviating from normality. The statistical outcomes were represented as mean ± SEM, considering *p* < 0.05 to indicate a notable difference and *p* < 0.01 to denote an exceedingly significant disparity.

## 3. Results

### 3.1. Physiologic Indexes

The impact of transportation on the physiological indexes of goats is presented in Table 1. In comparison to pre-transportation levels, after transportation, the body temperature, pulse, and respiratory frequency of goats increased very significantly (*p* < 0.01).

### 3.2. Rumen Fermentation Indexes

Table 2 illustrates the impact of transportation on the rumen fermentation indices of goats. In comparison to the pre-transportation period, there was a notable decrease in the pH value of the goats’ rumen after transportation (*p* < 0.01). Additionally, there was a significant increase in propionic acid content (*p* < 0.05), and the contents of TSCFA, acetic acid, butyric acid, isobutyric acid, isovaleric acid, HIS, and LPS exceedingly significantly increased (*p* < 0.01).

### 3.3. Blood Indexes

#### 3.3.1. Blood Biochemical Index

The impact of transportation on the blood biochemical indexes of goats is presented in Table 3. The analysis of blood biochemical-related indicators indicated that, in comparison to the pre-transportation period, the concentrations of ALT, AST, UREA, CREA, LDH, CK, LPS, and HIS increased very significantly after transportation (*p* < 0.01); conversely, the levels of TC, TG, and LDL-C decreased very notably (*p* < 0.01), and the ALP content significantly decreased (*p* < 0.05).

#### 3.3.2. Serum Immune Indexes

The impact of transportation on the serum immune indexes of goats is presented in Table 4. In comparison to the pre-transportation period, the contents of IL-1β, IL-6, TNF-α, MAP, and HSP-70 increased very significantly after transportation (*p* < 0.01). Conversely, the content of IgM, IgA, and IgG decreased very significantly (*p* < 0.01).

#### 3.3.3. Serum antioxidant indexes

The impact of transportation on the serum antioxidant indexes of goats is presented in Table 5. Compared to the pre-transportation period, there was a significant decrease in GPX content (*p* < 0.01) and a notable increase in protein carbonyl content (*p* < 0.01) after transportation.

#### 3.3.4. Serum Hormonal Indexes

The impact of transportation on the serum hormone indices of goats is detailed in Table 6. After transportation, there was a very significant decrease in the T4 content of serum hormones in goats (*p* < 0.01), while the levels of COR increased very notably (*p* < 0.01) and the EPI content showed a significant elevation (*p* < 0.05).

#### 3.3.5. Blood Routine Indexes

The impact of transportation on the blood routine indices of goats is presented in Table 7. In comparison to the pre-transportation period, there was an exceedingly notable decrease in the MID content of the blood routine indices of goats after transportation (*p* < 0.01). Conversely, the levels of HCT and RBC increased very significantly (*p* < 0.01), and the Hb content showed a significant elevation (*p* < 0.05).

### 3.4. Statistics of 16SrRNA Sequencing Results of Rumen Microorganisms

#### 3.4.1. Amplicon Sequence Variant Analysis

Specifically, it can be seen that the number of ASVs shared between the two treatment groups is 2096 (refer to Figure 1).

#### 3.4.2. Alpha Diversity Analysis

The results of the alpha diversity index analysis are presented in Figure 2. The results showed that there were no significant differences in Chao 1 (*p* = 0.96), Dominance (*p* = 0.37), Observed_otus (*p* = 0.98), Pielou_e (*p* = 0.80), Shannon (*p* = 0.84), and Simpson (*p* = 0.37) between the groups. This suggested that transportation had no impact on the richness and diversity of microbial communities.

#### 3.4.3. Beta Diversity Analysis

The findings are illustrated in Figure 3. Notably, only a handful of samples exhibited deviation within the two groups, while the remaining samples demonstrated relatively close proximity, lacking any discernible separation. These results showed that there were no substantial disparities in the structural composition of microorganisms between the two groups, indicating that transportation did not alter the structural makeup of rumen microorganisms in goats.

### 3.5. Analysis of Inter-Group Differences in Rumen Microorganisms

By referencing the annotation outcomes of ASVs and the attribute table of each sample, we derived the species abundance table at the levels of domain, phylum, class, order, family, genus, and species.

At the phylum level, we handpicked the top 10 microbial phyla based on their relative abundance in this sequencing for visual representation in a histogram (refer to Figure 4). Notably, Bacteroidota, Firmicutes, and Proteobacteria emerged as the predominant bacteria. Furthermore, we screened for bacterial phyla with an average relative abundance exceeding 0.1% in at least one group, considering them as major bacterial phyla. This criterion led to the identification of a total of 10 bacterial phyla (detailed in Table 8). Subsequently, we analyzed the disparities in relative abundance among the groups.

At the genus level, a histogram was made with the top 10 microbial genera with an average relative abundance (refer to Figure 5). Among these, *Succinivibricnaceae_UCG-001*, *Prevotella*, and Succinivibrio emerged as the most dominant genera. We considered bacterial genera with an average relative abundance exceeding 1% in at least one group as the primary bacterial genera. Following this criterion, a total of 10 bacterial genera were identified (detailed in Table 8). Subsequently, an analysis was conducted to assess the variations in relative abundance among the groups. Notably, no statistically significant differences were observed in the relative abundance of rumen flora before and after transportation, both at the phylum and genus levels (*p* > 0.05). It showed that transportation had no significant effect on rumen flora.

## 4. Discussion

Detection of serum cortisol and thyroid hormone concentrations is often used as a measure of the degree of stress in animals [9,10]. For ruminants, transport is also a strong stressor that activates the sympathetic nervous system and the hypothalamic–pituitary–adrenal axis, leading to an increase in serum cortisol concentrations [11]. Tajik et al. demonstrated that a mere 3 h transport resulted in a substantial 100% surge in serum cortisol levels among Iranian cashmere goats, with this increase persisting over the subsequent 24 h period [12]. Similarly, Fazio et al. revealed that transport stress induced a significant increase in thyroid and adrenal activity in cattle, which was evident even after short-distance transport and continued to increase after long-distance transport [13]. In this experiment, we observed a significant rise in serum cortisol and adrenaline levels among goats after transportation, which is consistent with the results of previous studies and also indicates that transport caused the stress response in goats in this experiment.

Physiological parameters such as pulse rate, body temperature, and respiratory rate change significantly when animals are subjected to stress [14]. In our experiment, compared with before transportation, the body temperature, respiratory rate, and pulse rate of goats increased significantly after transportation. Dixit et al. [15] observed similar findings in beef cattle subjected to both short-term (30 min) and long-term (14 h) transportation, where a significant elevation in body temperature was reported. This alignment with our findings suggests a consistent physiological response to transport stress across species. The reason for the increase in body temperature, respiratory rate, and pulse rate of goats after transportation may be due to the underdeveloped sweat glands of goats and their limited ability to dissipate heat through increased epidermal evaporation. During prolonged transportation, goats experience a prolonged stress state, leading to enhanced energy metabolism and increased heat production. As a result, the additional heat generated by transport stress can only be effectively dissipated through an elevated respiratory rate [16]. Over time, if the animal is unable to adequately dissipate this excess heat through respiratory and epidermal means, it will experience an increase in body temperature and a subsequent acceleration in pulse rate [17].

Serum markers such as T-AOC, GSH-PX, and other indicators are often used to measure the redox state of the organism [18,19,20,21]. Hosseinian et al. [22] found that after 6, 24, and 72 h of transportation, compared with the control group, the level of MDA in the serum of pigeons increased significantly, the T-AOC decreased significantly, and oxidative stress occurred in the body. Similarly, in our experiment, goats experienced a notable decrease in serum GSH-Px activity and a significant rise in protein carbonyl content following transportation, indicating that they, too, underwent oxidative stress. However, there are also inconsistent reports. Erdem et al. [23] reported a significant enhancement in serum GSH-Px and SOD (Superoxide Dismutase) activities in ewes after 4 h transport at 80 km/h. The reason for this discrepancy may be related to the different transportation time. It is plausible that short-term transportation might trigger a compensatory enhancement of antioxidant enzyme activity within a sheep’s body. With the continuous increase in transportation time, this compensatory mechanism may become exhausted, leading to a substantial depletion of antioxidant enzymes and an inevitable decline in the body’s total antioxidant capacity.

AST (Aspartate Aminotransferase) and ALT (Alanine Aminotransferase) serve as essential biomarkers for assessing liver health, with elevated ratios indicating more severe liver damage [24]. Our experimental findings revealed a substantial increase in both AST and ALT levels, along with a heightened ratio, following the transportation of goats. This suggests that the transportation process may have inflicted a certain degree of liver damage on them. Furthermore, urea and creatinine are nitrogen-containing end products of metabolism and are clinically used as screening indicators of renal function. Urea is the main metabolite derived from dietary protein and tissue protein conversion. Creatinine is the product of muscle creatine catabolism [25]. In line with previous studies by López Olvera et al. [26], which reported significant elevations in creatinine and urea levels in antelopes after transportation, our results also indicated potential kidney damage. Similarly, Dalmau et al. [27] observed a notable rise in blood urea levels in lambs after 24 h of transport, further corroborating our findings and suggesting that transportation may indeed contribute to renal impairment.

CK (Creatine Kinase) and LDH (Lactate Dehydrogenase) are important indicators reflecting the body’s energy metabolism in the bloodstream. When the body experiences continuous external stimuli, it can disrupt the function and permeability of the muscle cell membrane, resulting in the infiltration of CK and LDH in the cells to the outside world, causing the activity of CK and LDH in the serum to increase [28,29]. Shaw et al. [30] reported that stress significantly increased the serum CK activity of lambs, and the degree of muscle damage in lambs from different positions was also significantly different. This is consistent with the results of our experiment.

In animals undergoing stress, there is an augmentation of sympathetic nerve excitability coupled with a diminution in parasympathetic nerve excitability; this imbalance leads to the suppression of the gastrointestinal tract’s digestion and absorption capabilities. Consequently, in our experiment, we observed a notable elevation in rumen SCFA (short-chain fatty acid) concentration following transportation. The accumulation of SCFAs in the rumen will significantly lower the pH value of the rumen, and a large number of Gram-negative bacteria in the rumen will die in a low-pH environment [31], which in turn leads to the release of large amounts of LPS, resulting in a significant increase in rumen LPS concentration after transportation in this experiment. LPS, a primary constituent of the cell wall of Gram-negative bacteria, initiates an inflammatory cascade when it enters the body. It is recognized by pattern recognition receptors, leading to the release of pro-inflammatory factors such as TNF-α, IL-1β, and IL-6, as well as chemokines and other inflammatory mediators. This cascade results in a significant accumulation of neutrophils in the lungs, which collaborate with alveolar epithelial cells, vascular endothelial cells, and macrophages to further amplify the inflammatory response by releasing additional inflammatory mediators [32]. In our experiment, we found significant increases in the serum concentrations of LPS, IL-1β, IL-6, and TNF-α in goats after transportation. These findings suggested that transportation stress induces an inflammatory response in goats, aligning with previous studies conducted on beef cattle [7]. In the process of transportation, the large secretion of adrenal cortical hormones will affect the immune system and produce immunosuppressive effects [33]. Elevated levels of pro-inflammatory cytokines such as IL-1β, IL-6, and TNF-α can also lead to decreased immunity in animals [34,35]. According to Mao et al., the serum IgA level in Simmental cattle decreased notably after transportation [36]. Similarly, our experiment revealed a significant decrease in IgG, IgA, and IgM levels in goat serum following transportation, further corroborating that transportation-induced stress can impede normal immune function. In order to compensate for the decline in immunity, the body usually increases the synthesis of HSPs, which play an important role in protecting organisms against environmental stress. Hu et al. reported that the expression of HSP-70 in the stomach tissue of goats after transportation was significantly increased [4]. This experiment also found that the content of HSP-70 in the serum of goats after transportation was significantly increased.

Rumen microorganisms play an important role in the digestion of animal feed and the maintenance of body function [37]. Previous studies in cattle have shown that the OTU, Chao 1 and Shannon indices of rumen microbes were higher after transport than before transport [8], and the abundance of rumen bacteria was also significantly higher than before transport [38]. However, in our experiment involving goats as the test subjects, we observed no notable differences in either the α- or β-diversity of rumen microorganisms before and after transportation. Additionally, there were no significant disparities in the relative flora abundance between groups at the phylum and genus levels, indicating that transportation had a minimal impact on the rumen bacterial structure and composition in goats. This variation could be attributed to the distinct rumen microbial communities present in cattle and goats [39,40]. According to Li et al. [41], the rumen microbiota composition differs significantly between these two animal species. At the phylum level, Firmicutes dominates in cattle, whereas Proteobacteria was the prevalent phylum in goats. Similarly, at the genus level, Prevotella and Streptococcus were the main dominant genera in cattle, while Escherichia–Shigella was the main dominant genus in goats. Our findings, which show no substantial changes in the structural composition of the rumen microbiota in goats pre- and post-transportation, suggest that the rumen microbiota of goats possesses a remarkable adaptive capacity. Despite the intense stress experienced by the goat during transportation, the rumen microflora appears to have acclimated by adjusting its metabolism.

## 5. Conclusions

Continuous transportation for a duration of 20 h can induce a severe stress response in goats, leading to compromised immune function, diminished antioxidant capacity, escalated inflammatory response, and altered rumen fermentation indices. However, the experiment did not reveal any significant impact on the structure and composition of the rumen microbiota.

## Figures and Tables

**Figure 1 animals-14-01616-f001:**
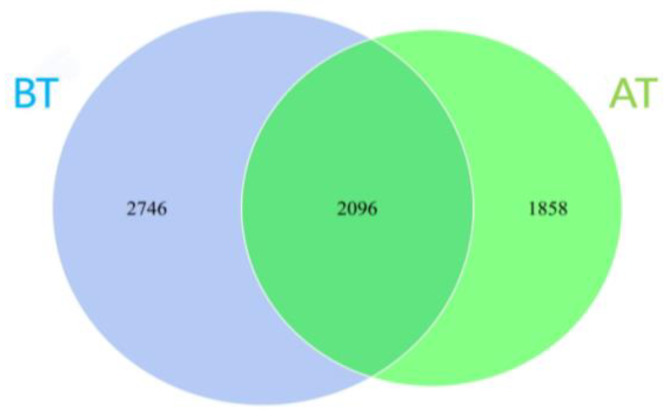
The Venn plot of ASV.

**Figure 2 animals-14-01616-f002:**
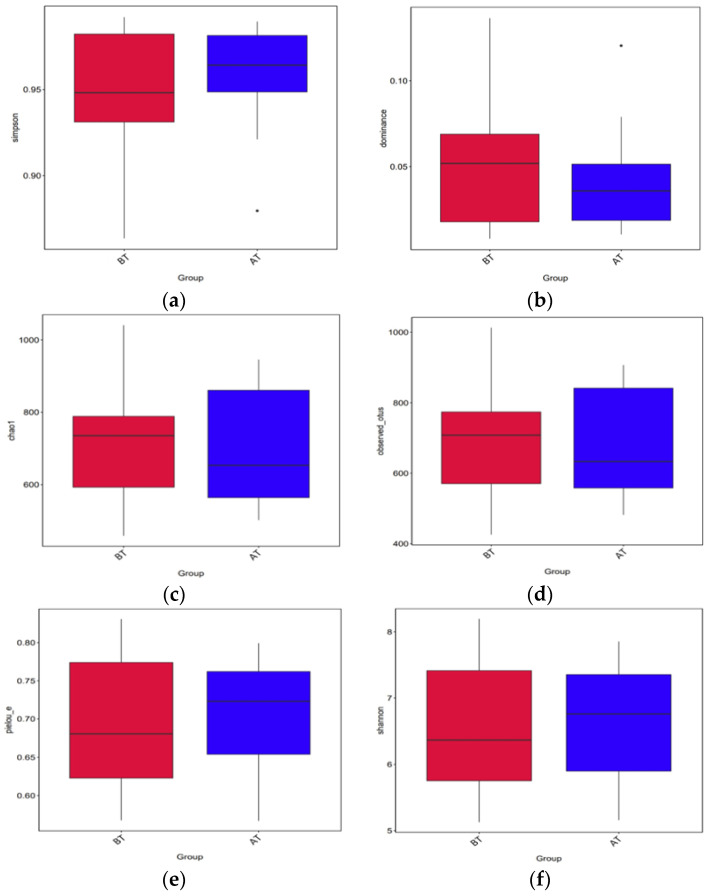
Analysis of microbial alpha diversity between groups. The read color means before transportation and the blue color means after transportation. BT: before transportation; AT: after transportation; (**a**) is the comparison of Simpson index, (**b**) is the comparison of Dominance index, (**c**) is the comparison of Chao 1 index, (**d**) is the comparison of Observed otus index, (**e**) is the comparison of Pielou_e index, (**f**) is the result of Shannon index.

**Figure 3 animals-14-01616-f003:**
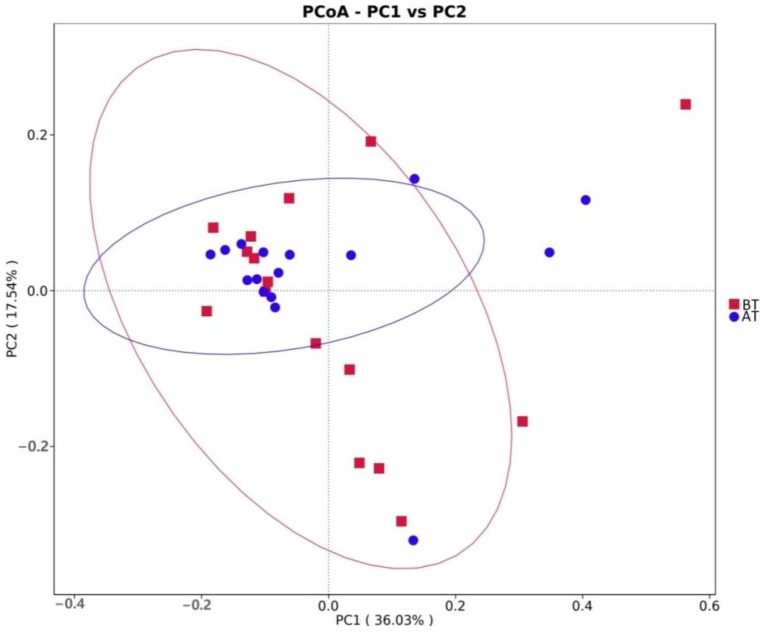
PCoA map of rumen microorganisms.BT: before transportation; AT: after transportation; Read line: before transportation; Blue line: after transportation.

**Figure 4 animals-14-01616-f004:**
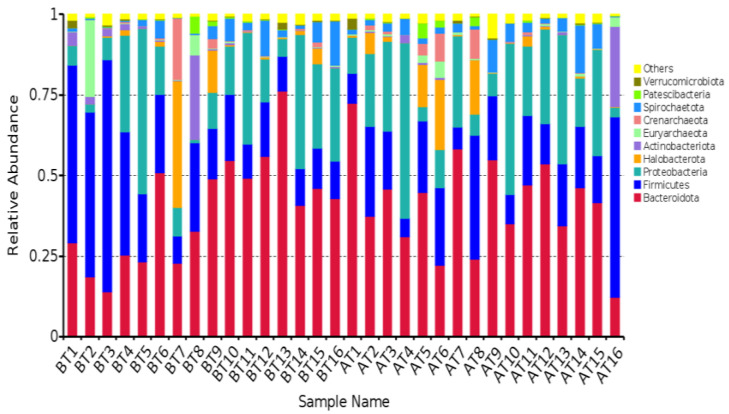
The species composition of the dominant microbiota at the phylum level (Top 10).

**Figure 5 animals-14-01616-f005:**
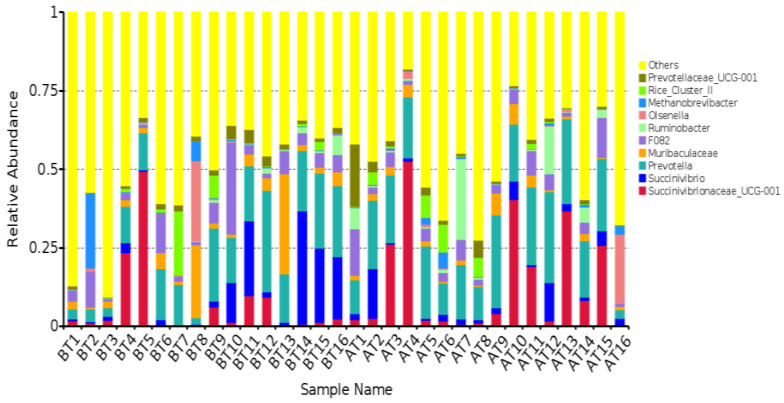
The species composition of the dominant microbiota at the genus level (top 10).

**Table 1 animals-14-01616-t001:** Effect of transportation on physiological indexes of goats.

Item	BT	AT	SEM	*p*-Value
Body temperature (°C)	38.65	39.46	0.05	<0.01
Pulse (times/min)	97.79	107.58	2.89	<0.01
Respiratory frequency (times/min)	33.68	39.05	0.47	<0.01

BT: before transportation; AT: after transportation.

**Table 2 animals-14-01616-t002:** Effect of transportation on rumen fermentation indexes in goats.

Item	BT	AT	SEM	*p*-Value
pH Value	7.05	6.78	0.04	<0.01
TSCFA (mmol/L)	39.89	70.19	3.90	<0.01
Acetic acid (mmol/L)	25.18	43.72	2.32	<0.01
Propionic acid (mmol/L)	10.68	19.57	1.39	<0.05
Acetic acid/Propionic acid	2.69	2.31	0.16	0.60
Butyric acid (mmol/L)	2.21	4.37	0.27	<0.01
Isobutyric acid (mmol/L)	0.60	0.90	0.04	<0.01
Valeric acid (mmol/L)	0.42	0.72	0.06	0.05
Isovaleric acid (mmol/L)	0.83	1.25	0.05	<0.01
Microprotein (μg/L)	24.45	33.23	2.34	0.11
NH3-N (mg/dL)	65.92	56.31	3.22	0.16
HIS (ng/mL)	3.40	4.26	0.12	<0.01
LPS (ng/L)	71.84	95.74	3.19	<0.01

BT: before transportation; AT: after transportation; HIS: Histamine; LPS: Lipopolysaccharide.

**Table 3 animals-14-01616-t003:** Effect of transportation on blood biochemical indexes in goats.

Item	BT	AT	SEM	*p*-Value
ALT (U/L)	20.92	26.51	0.85	<0.01
AST (U/L)	67.30	87.11	3.33	<0.01
UREA (mmol/L)	6.40	8.64	0.28	<0.01
CREA (µmol/L)	60.66	86.31	2.89	<0.01
ALP (U/L)	389.20	207.66	46.23	<0.05
LDH (U/L)	288.78	419.38	16.50	<0.01
CK (U/L)	110.10	298.76	19.91	<0.01
TP (g/L)	56.43	559.79	1.40	0.42
Alb (g/L)	27.96	30.88	0.80	0.13
TC (mmol/L)	1.55	1.23	0.06	<0.01
TG (mmol/L)	0.32	0.15	0.02	<0.01
LDL-C (mmol/L)	0.56	0.37	0.03	<0.01
HDL-C (mmol/L)	1.52	1.41	0.05	0.30
LPS (ng/L)	251.66	458.12	19.74	<0.01
HIS (ng/mL)	9.93	16.54	0.53	<0.01

BT: before transportation; AT: after transportation; ALT: Alanine Aminotransferase; CREA: creatinine; AST: Aspartate Aminotransferase; ALP: Alkaline Phosphatase; LDH: Lactic Dehydrogenase; CK: Creatine Kinase; TP: Total Protein; Alb: Albumin; TC: Total Cholesterol; TG: Triglyceride; HIS: Histamine; LPS: Lipopolysaccharide; LDL-C: Low-Density Lipoprotein Cholesterol; HDL-C: High-Density Lipoprotein Cholesterol.

**Table 4 animals-14-01616-t004:** Effect of transportation on serum immune indexes in goats.

Item	BT	AT	SEM	*p*-Value
IL-1β (pg/mL)	49.57	94.42	4.91	<0.01
IL-6 (pg/mL)	56.84	99.90	5.17	<0.01
IL-10 (pg/mL)	35.99	34.50	1.36	0.60
TNF-α (pg/mL)	115.60	176.05	6.81	<0.01
IgM (μg/mL)	1904.12	1299.39	58.28	<0.01
IgA (μg/mL)	186.32	130.39	8.01	<0.01
IgG (mg/mL)	6.59	4.83	0.21	<0.01
MAP (μg/mL)	105.90	168.35	5.91	<0.01
HSP-70 (pg/mL)	343.35	537.96	25.13	<0.01

BT: before transportation; AT: after transportation; MAP: Major Acute Phase Protein; HSP-70: Heat Shock Protein-70.

**Table 5 animals-14-01616-t005:** Effect of transportation on serum antioxidation indexes in goats.

Item	BT	AT	SEM	*p*-Value
MDA (nmol/mL)	2.28	1.80	0.17	0.14
T-AOC (μmol/mL)	0.12	0.11	0.006	0.42
GPX (U/mL)	36.18	21.28	1.50	<0.01
SOD (U/mL)	1.10	1.22	0.13	0.66
protein carbonyl (μmol/mL)	0.04	0.08	0.01	<0.01

BT: before transportation; AT: after transportation; MDA: Malondialdehyde; T-AOC: total antioxidant capacity; GPX: Glutathione Peroxidase; SOD: Superoxide Dismutase.

**Table 6 animals-14-01616-t006:** Effect of transportation on blood hormone indexes of goats.

Item	BT	AT	SEM	*p*-Value
T4 (ng/mL)	178.49	131.44	5.75	<0.01
COR (ng/mL)	63.80	100.02	3.35	<0.01
EPI (pg/mL)	1329.79	1545.93	46.68	<0.05

BT: before transportation; AT: after transportation; T4: Thyroxine; COR: cortisol; EPI: Epinephrine.

**Table 7 animals-14-01616-t007:** Effect of transportation on blood routine indexes of goats.

Item	BT	AT	SEM	*p*-Value
WBC (×10^9^/L)	14.97	15.91	0.85	0.40
NEU (%)	16.41	21.70	3.02	0.22
LYM (%)	73.38	71.97	3.53	0.77
MID (%)	1.15	1.05	0.30	<0.01
HCT (%)	8.77	11.06	0.42	<0.01
Hb (g/L)	114.45	120.21	1.95	<0.05
RBC (×10^12^/L)	1.96	3.01	0.09	<0.01

BT: before transportation; AT: after transportation; WBC: White Blood Cell; NEU: Neutrophilic Granulocyte; LYM: Lymphocyte; MID: Intermediate Cell; HCT: Hematocrit; Hb: Hemoglobin; RBC: Red Blood Cell.

**Table 8 animals-14-01616-t008:** Analysis of inter-group differences in the relative abundance of dominant rumen microorganisms at the phylum and genus levels (top 10).

Item	Groups		*p*-Value
	BT	AT	
Phylum			
Bacteroidetes	39.44 ± 4.19	41.31 ± 3.76	0.74
Spirochaetota	0.04 ± 0.01	0.04 ± 0.0 1	0.75
Firmicutes	25.51 ± 4.73	20.26 ± 3.18	0.36
Proteobacteria	18.62 ± 3.75	22.71 ± 3.90	0.46
Actinobacteriota	2.51 ± 1.60	1.93 ± 1.53	0.80
Halobacterota	4.00 ± 2.49	4.13 ± 1.73	0.96
Euryarchaeota	2.06 ± 1.51	0.93 ± 0.34	0.47
Crenarchaeota	1.61 ± 1.18	1.58 ± 0.75	0.99
Patescibacteria	0.59 ± 0.32	0.70 ± 0.32	0.80
Verrucomicrobiota	0.40 ± 0.17	0.41 ± 0.21	0.96
Others	1.65 ± 0.20	1.95 ± 0.40	0.51
genus			
*Olsenella*	1.81 ± 1.59	1.70 ± 1.37	0.96
*Prevotella*	15.61 ± 2.17	19.15 ± 1.88	0.13
*Muribaculaceae*	5.55 ± 2.20	2.31 ± 0.51	0.17
*F082*	6.02 ± 1.78	4.78 ± 1.01	0.55
*Methanobrevibacter*	2.05 ± 1.51	0.91 ± 0.34	0.47
*Succinivibrio*	8.05 ± 2.87	3.50 ± 1.11	0.16
*Succinivibrionaceae_UCG-001*	6.81 ± 3.21	14.05 ± 4.28	0.19
*Ruminobacter*	0.79 ± 0.39	3.76 ± 1.76	0.12
*Prevotellaceae_UCG-001*	1.67 ± 0.31	2.61 ± 1.20	0.45
*Rice_Cluster_II*	2.13 ± 1.30	1.83 ± 0.73	0.84
Others	50.51 ± 4.43	45.40 ± 4.05	0.40

BT: before transportation; AT: after transportation.

## Data Availability

The data that support the findings in this study were not deposited in an official repository. These data are available from the authors upon request.

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
