# Peer review of "Effects of Transportation on Blood Indices, Oxidative Stress, Rumen Fermentation Parameters and Rumen Microbiota in Goats"

_animals, 2024, doi:10.3390/ani14111616_

Round 1
Reviewer 1 Report
Comments and Suggestions for Authors
Which protocol was adapted for concentration of VFA?
Referencing can be improved (numbering of some references were omitted).
Comments on the Quality of English Language
Words construction and grammatical errors (for example: exceedingly significantly). The result section and discussion can be improved.
Author Response
We gratefully appreciate for your suggestions and thank you very much for your suggestion and professional advice.These suggestions are very helpful to my article. According to your suggestion, we have revised the manuscript of the article. The details as follows:
- In the part of 2.3.2 ( 3 ), we described the measurement of VFAs concentration, the concentration of VFAs was ascertained by gas chromatography (CP-3800, Varian).
- Thank you very much for your advice, we have improved the references.
- Thank you for your reminder, we are so sorry for our incorrect writing and we have already changed the "exceedingly significantly" in 3.2 to "exceddingly significant " and we have already revised our manuscripts.
Thank you again for your valuable comments and suggestions on our manuscript. We would be glad to respond to any questions and comments that you may have.
Reviewer 2 Report
Comments and Suggestions for Authors
The paper provided interesting and insightful information on "Effects of Transportation on Blood Indices, Oxidative Stress, Rumen Fermentation Parameters and Rumen Microbial Flora in Goats", but there are many small mistakes regarding punctuation, grammar, word type of unit, and some others. Please carefully check these mistakes thoroughly. In addition, the following obvious errors need to be corrected and improved:
1. Abstract:Non-generic abbreviations that appear for the first time should have full names. The same is required in the text.
2. Materials and Methods:
L 66-67: The composition and nutrient level of the diet should be shown in a table.
3. Results:
Table 2: “umol/L” should be changed to “µmol/L”
Table 6: “WBC (109/L)” and “RBC(1012/L)” should be changed as“WBC (×109/L)” and “RBC(×1012/L)”
Table 7: “t/min” should be changed to “times/min”
Comments on the Quality of English Language
There are many small mistakes regarding punctuation, grammar, word type of unit, and some others.
Author Response
We gratefully appreciate for your suggestions and thank you very much for your suggestion and professional advice.These suggestions are very helpful to my article. According to your suggestion, we have revised the manuscript of the article. The details as follows:
- I have renamed the first non-universal acronym in the article to full name;
- Thank you for your advice, we have carefully considered it, as transport stress is part of our group trials, in order to avoid duplication with other articles to be published, and our research focus is mainly on transport stress, with little effect on feed formulations ; so we chose to show the dite composition in this wayï¼›
- Thank you for your reminder, we are so sorry for our incorrect writing and the “umol/L” had been changed to “µmol/L” and presented in Table 3;
- The “WBC (109/L)” and “RBC(1012/L)” had been changed to“WBC (×109/L)” and “RBC(×1012/L)” and presented in Table 7;
- The “t/min” had been changed to “times/min” and presented in Table 1.
Thank you again for giving us the opportunity to strengthen the manuscript with your valuable comments. If there are any other modifications we could make, we would like very much to modify them and we really appreciate your help.
Reviewer 3 Report
Comments and Suggestions for Authors
Dear authors,
The manuscript (animals-3004298) investigates the influences of 20-hour transportation on blood indices, oxidative stress, rumen fermentation parameters and rumen microbial structure and composition in goats. The content of this manuscript totally falls within the scope of animals. To the best of my knowledge this paper has not been published elsewhere.
The authors conducted a complicated experiment on 16 goats with a wide range of measurements including physiological indexes, blood and serum parameters, rumen fermentation attributes, rumen microorganism composition... The manuscript is relatively well-prepared with a clear research objective. After carefully reviewing the manuscript, I suggest that it will be suitable for publication in this journal with a major revision. I have several comments for the authors to consider as follows:
The Simple Summary section was mainly physically copied and pasted from the other parts of the paper. Please try to paraphrase the sentences.
Abstract: remove p values in Abstract.
Keywords: In my opinion, please try to avoid words which appeared in the title.
Materials and Methods:
It is recommended to provide goat breed if applicable.
Please provide some reference or explain why goat were transported on days 17 and 18?
Provide more detail about the vehicle: type of vehicle, floor surface, crate (container) dimensions, how may decks, ventilation system…
Between blood and rumen liquid sample which ones were collected first and why? Please mention in the manuscript.
Results:
3.2.6. Physiologic indexes: move this part to the beginning of the result section.
Lines 274-278: move these sentences up to the Materials and Methods section.
Lines 287-289: as above.
Discussion
Remove all initials of the in-text citations. Please follow journal citation style requirement.
Lines 287-289: need a reference.
Comments on the Quality of English Language
Some sentences are grammatically incorrect. Use the full form of the words in the subheading and at the first time following with their abbreviation.
Author Response
We gratefully appreciate for your suggestions and thank you very much for your suggestion and professional advice.These suggestions are very helpful to my article. According to your suggestion, we have revised the manuscript of the article. The details as follows:
- According to your suggestion, we partially modified the simple summary. In this part of the simple summary, we first described the importance of transportation ( transportation is an inevitable part of the goat raising process ). However, transportation could lead to transportation stress. Therefore, in order to understand the impact of transportation stress on goats, we conducted our experiment. Then, we briefly described our experimental design and experimental results, and got the experimental conclusion.
- We have removed p values in Abstract.
- Thank you for your suggestion. Your suggestion is very good. We deleted the keyword of oxidative stress. However, since this article mainly studies the transport stress of goats, some keywords such as goats, transport stress, rumen fermentation, rumen microbiota are closely related to our article, so we are so sorry that it is difficult for us to modify themï¼›
- The breed of goat is JianZhou Big-Eared Goat and we have added it in the 2.1 according to your suggestion.
- According to our previous breeding experience, in the process of breeding, it usually takes about 2 weeks for the animals to adapt to the feed and the environment. Therefore, our experiment designed a 16-day feeding experiment and began transportation in on days 17 and 18.
- We used a fence type truck, single layer, high fence, natural ventilation, size of 4.2 m long, 2.1 m wide, 2.1 m high.
- Between blood and rumen fluid sample, we first collectd the blood ; because it took a relatively short time to draw blood compared with rumen fluid, the stress produced by animals may be small, and the blood is more sensitive to stress than rumen fluid.
- We have moved Physiologic indexes to the beginning of the result section.
- According to your suggestion,we have moved lines 274-278 and lines 287-289 up to the Materials and Methods section.
- We have removed all initials of the in-text citation in dissicuion.
Finally, thank you again for your time to review our manuscript. If there are any other modifications we could make, we would like very much to modify them and we really appreciate your help.
Round 2
Reviewer 3 Report
Comments and Suggestions for Authors
Dear Authors,
I am happy with your response. In my opinion, the paper could be accepted after revising some grammatic and typo errors.
Kind regards,
Comments on the Quality of English Language
Some grammatic and typo errors need to be corrected.